# Strain-resolved analysis of hospital rooms and infants reveals overlap between the human and room microbiome

Brandon Brooks [1], Matthew R. Olm[1], Brian A. Firek[2], Robyn Baker[3], Brian C. Thomas[4], Michael J. Morowitz[2] & Jillian F. Banfield[4]

Preterm infants exhibit different microbiome colonization patterns relative to full-term infants, and it is speculated that the hospital room environment may contribute to infant microbiome development. Here, we present a genome-resolved metagenomic study of microbial genotypes from the gastrointestinal tracts of infants and from the neonatal intensive care unit (NICU) room environment. Some strains detected in hospitalized infants also occur in sinks and on surfaces, and belong to species such as *Staphylococcus epidermidis*, *Enterococcus faecalis*, *Pseudomonas aeruginosa*, and *Klebsiella pneumoniae*, which are frequently implicated in nosocomial infection and preterm infant gut colonization. Of the 15 *K. pneumoniae* strains detected in the study, four were detected in both infant gut and room samples. Time series experiments showed that nearly all strains associated with infant gut colonization can be detected in the room after, and often before, detection in the gut. Thus, we conclude that a component of premature infant gut colonization is the cycle of microbial exchange between the room and the occupant.

[1] Department of Plant and Microbial Biology, University of California, Berkeley, CA 94720, USA. [2] Department of Surgery, University of Pittsburgh School of Medicine, Pittsburgh, PA 15261, USA. [3] Division of Newborn Medicine, Magee-Womens Hospital of Pittsburgh of UPMC, Pittsburgh, PA 15224, USA. [4] Department of Earth and Planetary Sciences, and Environmental Science, Policy and Management, University of California, Berkeley, CA 94720, USA. Correspondence and requests for materials should be addressed to J.F.B. (email: jbanfield@berkeley.edu)

The sources of microbes that colonize newborn infants remain unclear. In the case of hospitalized premature infants, previous hospital microbiome studies[1, 2] have implicated the room environment as an important source of colonizing bacteria. However, relatively little is known about the overall microbiome composition of hospital rooms, the distributions of organisms across room locations, and the specific reservoirs of strains that could potentially colonize patients. Prior studies of hospital room microbiomes were conducted using 16S ribosomal RNA surveys that cannot reliably distinguish different bacterial species[3, 4]. This is important because different species and even strains of the same species can differ enormously in

their pathogenicity and antibiotic resistance[5]. Very detailed, genome-wide comparisons are needed to determine if a hospital room is the source of organisms detected in infants or if organisms were introduced to the infant from external sources. This can be accomplished by genomic sequencing of isolates, but this targeted approach does not address the broader question of overall community composition of hospital room environments.

Here, we conduct a genome-resolved metagenomic study targeting surfaces and sinks of hospital rooms and fecal samples of premature infants hospitalized in those rooms. We use genome-to-genome comparisons to identify organisms in the hospital room and infants and establish direct links between strains in specific neonatal intensive care unit (NICU) environments and the infant gut. The analysis involves metagenomic data from pooled low-biomass samples from NICU surfaces and a terabase-scale metagenomic data set representing the gastrointestinal tracts of 50 premature infants. We identify reservoirs of bacteria from groups known to colonize the infants and show that some specific strains detected in hospitalized infants also occur in sinks and on surfaces. Thus, we provide evidence for the cycle of room—infant strain exchange.

## Results

**Sampling and sequencing.** We collected 1038 samples from NICU rooms housing six premature infants (S2_2013 cohort; Supplementary Fig. 1). Room samples were derived from a variety of touched surfaces, as well as from the sink basin interiors. Due to extremely low biomass, DNA from multiple samples collected at different times from the same room environment type was pooled, generating three sample types per room: swabs from frequently touched surfaces, wipes from other surfaces, and swabs from sinks (Supplementary Data 1).

To enable genomic comparison of strains present in the NICU environment with infant-associated strains, samples were collected from newborn, almost exclusively preterm infants housed in the NICU of the Magee-Womens' Hospital of UPMC, Pittsburgh (USA). Our analyses made use of 425 newly acquired metagenomes for 29 infants, as well as two previously published data sets for 21 infants[6, 7] (Supplementary Data 2). In total, we analyzed 622 metagenomes from 50 infants that were sampled over an ~3-year period (see Supplementary Fig. 1 for an experimental overview). The earliest fecal samples were collected on day of life (DOL) 5 and the latest collection point was DOL 86.

DNA from both room and fecal samples was sequenced, generating 2.34 Tb of new sequence information. Sequencing reads were trimmed, assembled, and the data were binned using a previously described metagenomic analysis pipeline[8]. For room samples, sequencing allocations were larger than those for fecal samples to compensate for their expected higher levels of microbial diversity.

**Successful genome reconstruction from room environments.** We generated hundreds of draft quality genomes for bacteria present in both the infant gut and room samples. The successful recovery of high-quality genomes directly from low-biomass, room-collected samples is unprecedented. In total, we reconstructed 131

unique (dereplicated) high-quality genomes for distinct room-associated strains and a total of 317 unique genomes for strains present in infant fecal samples.

**Organisms found in the infant gut have room reservoirs.** Prior genome-based metagenomic studies suggested that most infants housed in the same NICU harbor microbial taxa that are similar at the species level but distinct at the level of strain[6, 7]. An exception to this observation is a group of persistent strains that were observed in multiple infants, often several years apart[7]. In this prior study, a strain was considered the same if two near-complete genome bins had greater than 98% average nucleotide identity (ANI) across 95% of the bin. The results suggested that room reservoirs may exist for the "persister" strains, and that reseeding of sequential room occupants occurs from these reservoirs.

To address the question of potential room-associated reservoirs of bacteria from groups known to colonize the infant gut, we conducted a first analysis in which we defined "subspecies" clusters using dRep[9] with a whole-genome ANI (gANI)[10] of > 99% (Supplementary Data 3–11). This threshold is comparable to the stringency achieved through 98% ANI over the vast majority of the genome. Although it is not sufficient to establish that samples with the same subspecies cluster share a common source (which is tested for below), it enables identification of tightly defined populations that can be used to identify room reservoirs of bacteria that are likely human microbiome relevant.

Of the 317 "subspecies" detected in room samples, only 40 were detected in at least two samples, yet of the 131 subspecies defined from infants, 75 were found in two or more infants (Supplementary Data 3–11). Of particular interest are the 12 subspecies found in both room and infant samples (Fig. 1). The most commonly detected subspecies found in both fecal and room samples were, in order of decreasing frequency, *E. faecalis*, *S. epidermidis*, *K. pneumoniae*, *Propionibacterium avidum*, *Escherichia coli*, and *P. aeruginosa* (for details, see Supplementary Fig. 2). This is important because these are among the most common populations detected in the 50 infants studied here. Interestingly, subspecies that include three of the five previously reported "persister" strains (*E. coli*, *E. faecalis*, and *P. aeruginosa*) were present in room samples (Fig. 1 and Supplementary Fig. 2). Thus, we conclude that room habitats contained bacterial groups that often colonize infants housed in the NICU.

Interestingly, species of Clostridia, which are common infant gut colonizers, were very rare in the room habitats (< 1% of the community in each sample based on scaffold profiling). This observation may indicate that these anaerobes are not well suited to grow in sinks or on surfaces in the NICU, that they are effectively removed by cleaning practices, or sporulation-limited DNA recovery.

**Evidence of strain transfer among infants and rooms.** In a second analysis, we used the stringent cutoff of 99.999% ANI to compare genomes recovered from the room and infant gut and identify cases of direct strain transfer. This threshold was chosen based on analysis of read variation in samples from which genomes

**Fig. 1** Subspecies found in the infant gut across several cohorts and years are also present in rooms. Fecal (columns 1, 2, 3, and 5) and room (column 4) samples from defined sampling time periods (the 2011, 2012, 2013, and 2014 cohorts) are arrayed on the x axis. Dots indicate the percent of infants, or rooms for column 4, in each cohort that contain a given subspecies (the top three highest percentages are labeled for clarity). The y axis labels indicate a representative genotype from clusters of genotypes that share > 99% gANI. The first digit in the name indicates the species and the second indicates the specific subspecies. See Supplementary Data 3–11 for further genome-clustering results. Only genomes that were found across different cohorts or in both an infant and a room are displayed. Triangles indicate organisms classified as persistent taxa ("persisters") because they appear in the room and in infant cohorts more than 1 year apart

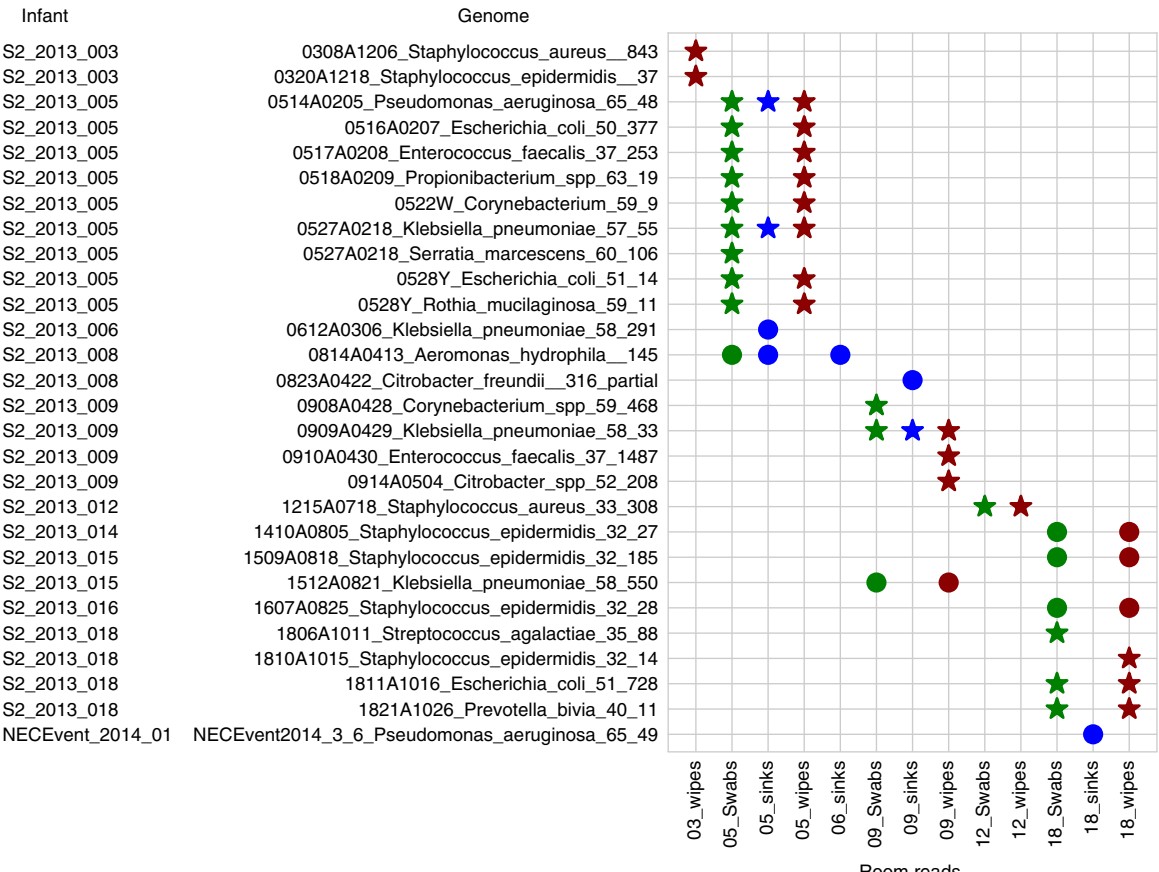

**Fig. 2** Gut strains are consistently detected in the room. Room reads from all room samples were mapped to each infant's dereplicated fecal genome set. Symbols indicate fecal genomes present in the room with at least 99.999% ANI. Circles represent cases where a fecal genome was detected in a separate room from which the infant was housed, and stars represent matched infant/room pairs. Wipe, swab, and sink metagenomes are filled in red, green, and blue, respectively

were assembled (essentially the limit of detection for the method, i.e., ~50 single-nucleotide polymorphisms (SNPs) per genome, Supplementary Fig. 3). This analysis revealed 28 cases of identical strains present in both the room environment and infant gut (rows in Fig. 2, Supplementary Data 12). In 20 of these cases, the room and infant samples were matched (the room housed the infant). Transferred strains were much more likely to be present in the swabs (n = 20) or wipes (n = 21) than in the sink samples (n = 7), indicating reservoirs on hospital surfaces. Fifteen of the 28 strains were present in both the wipe and swabs, and *K. pneumoniae* (a common cause of hospital-acquired infections) was present in all sample types.

Several interesting observations arise from the strain analysis (Fig. 2). Strain sharing occurred in infants hospitalized at the same time and infants separated by many months. The same *Aeromonas hydrophilia* (a potential pathogen) strain colonized infants 5 and 6's room samples and later colonized infant 8's gut (a 14-day separation). A *K. pneumoniae* strain from infant 9's gut samples was later found in infant 15's room (a 90-day separation). Interestingly, these room pairs are either adjacent to or across from one another, possibly indicating biogeographical localization within the NICU (for a map of the NICU, see previously published floorplan[11]). Surprisingly, five of the transferred strains were of the species *S. epidermidis*. Three of these strains were found in infants prior to and/or after they were found in the room environment. These findings may indicate regular carriage of this skin-associated bacterium via healthcare providers and emphasize the ability of *S. epidermidis* to exist both in the room environment and colonize the newborn infant gut.

Four *K. pneumoniae* strains were detected in both the room and infant gut environments. Two strains occurred in all sampled environments of the room that housed that infant (infants 5 and 9). A third strain was detected in the gut of infant 15 and in the room of infant 9. Similarly, the fourth strain colonized infant 6 and occurred in the sink of infant 5. Because the sink samples of infant 5 were collected prior to the birth of infant 6, and the swab and wipe samples from infant 9's room were collected prior to the birth of infant 15, the results indicate transfer from rooms to infants. In these documented cases, the infants acquired strains detected in other rooms, but the strains may be widely distributed in the NICU. Interestingly, the room-derived *K. pneumoniae* strain is transient in infant 6, and is replaced by DOL 20 by a very closely related strain (Supplementary Fig. 4).

**Temporal analysis involving one infant.** To further compare strains present in the room to those in infants, room samples were collected prior to, during, and after the hospitalization period of infant 5. This infant was chosen based on sample availability, as well as access to published microbial genomic data from this infant's mouth and skin[8]. The hospitalization period was subdivided into early (DOL 5–12), middle (DOL 13–20), and late (DOL 21–28) time intervals, and swabs, sinks, and wipes collected from these intervals were pooled to ensure sufficient DNA for sequencing from each environment type. The comparison was done at both the subspecies (99% ANI) and strain (99.999% ANI) level (Fig. 3 and Supplementary Fig. 4, respectively). Eleven of the 12 subspecies types associated with infant 5

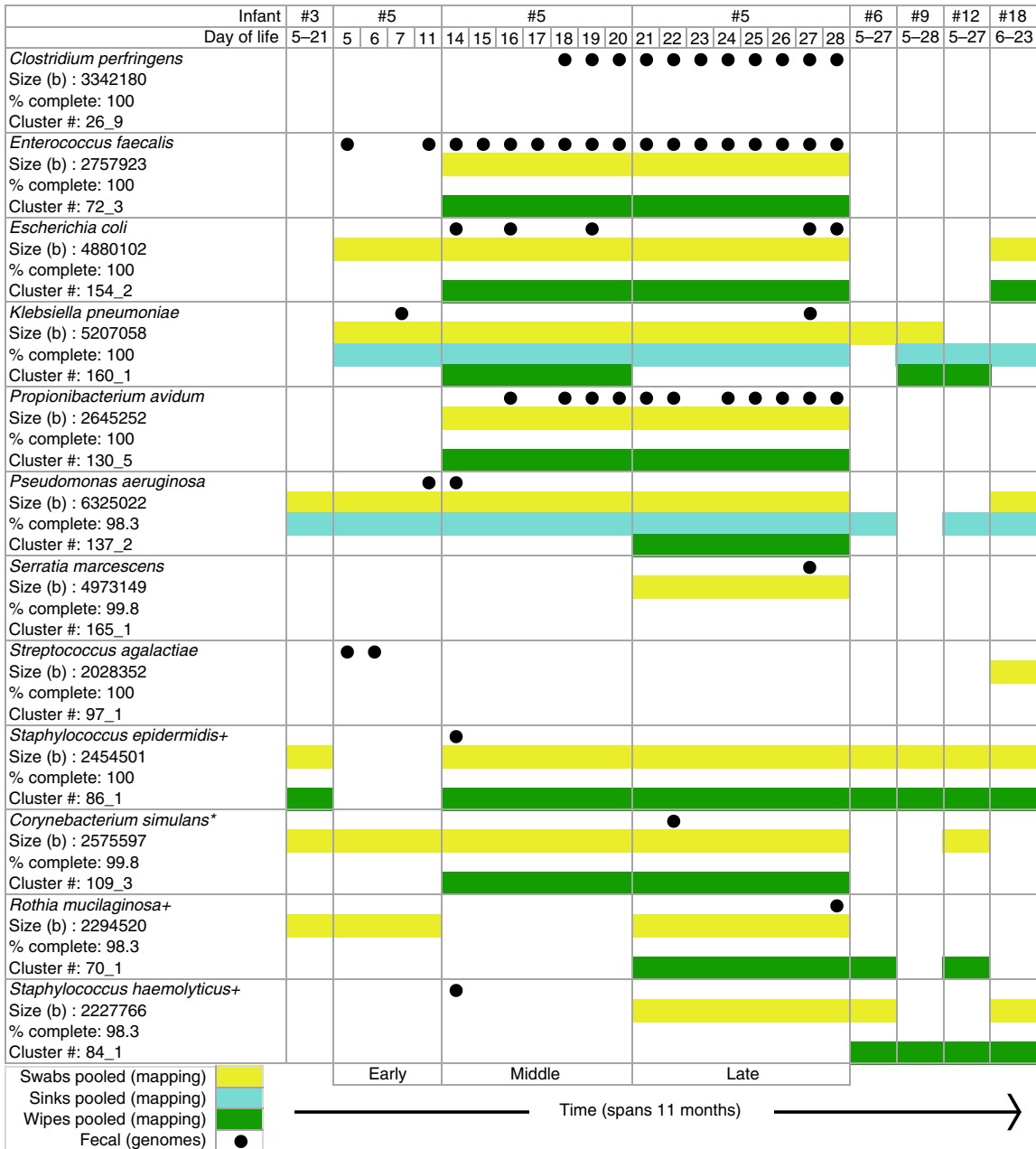

**Fig. 3** Strains associated with infant 5 also occurred in this infant's room and in rooms of previously and subsequently hospitalized infants. Detection in swabs from frequently touched surfaces (yellow), sinks (blue), and wipes from other surfaces (green). The results are shown in chronological order from left to right. Genomes recovered from specific days of life from infant 5's samples are indicated by black dots. Room detection of strains is indicated by a colored bar because room samples from one environment type were pooled, either by infant or for three time periods for infant 5

were detected in room samples, in some cases both before and after the hospitalization period of infant 5 (Fig. 3). *S. epidermidis*, *S. heamolyticus*, *P. aeruginosa*, and *K. pneumoniae* occurred in at least five of the six studied rooms. This result demonstrates that microbiome-relevant organisms are present in the NICU room environment over almost a 1-year study period. Nine strains co-occurred in infant 5 and its room; eight of these were detected in different sampling periods and occurred in two sample types. Overall, many infants were colonized by a strain that was closely related to, but distinct from the one detected in its room. For example, the specific strains of infant 5's gut were not detected in the rooms of other infants (Fig. 2). Thus, we conclude that although strain transfer among rooms and infants can occur, it is not the dominant process. It is possible, however, that deeper sequencing of the room would reveal the gut-associated

strains at low abundance and the predominance of the gut strain reflects strong genotypic selection from a complex room-associated strain reservoir.

## Discussion

Newborn preterm infants are commonly colonized by nosocomial antibiotic-resistant pathogens, and are at increased risk for serious infection and death[12]. The detailed genomic evidence presented here demonstrates that rooms can be reservoirs for early-stage colonizers of the microbiome of hospitalized premature infants. This finding provides a mechanism to explain how infants in the same NICU could be colonized by the same strains, despite hospitalization periods separated by years. We conclude that the room environment should be regarded along with diet,

mode of delivery, and antibiotics as a determinant of the early gut microbiome.

## Methods

**Sample collection, extraction, and sequencing**. All samples were collected after signed guardian consent was obtained, as outlined in our protocol to the ethical research board of the University of Pittsburgh (IRB PRO12100487 and PRO10090089). This consent included sample collection permissions and consent to publish study findings. Swabs, saturated with 0.15 M NaCl and 0.1% Tween20 sampling buffer, were collected Monday through Friday using previously described methods[1], the only change being the use of a nylon FLOQSwab (Copan Diagnostics, Brescia, Italy). Wipe samples were collected from the floor and exterior top of the isolette using Texwipe TX1086 wipes (Texwipe, Kernersville, NC, USA). Before collecting each wipe sample, the collector put on latex examination gloves and cleaned these gloves with an isopropanol wipe. The wiped surface area was approximately 48 cm$^2$ or, for smaller surfaces, the entire surface itself (e.g., isolette top). Fecal samples collected from infant diapers were kept frozen on dry ice and up to 0.25 g of thawed sample was added to tubes or wells of extraction blocks from the PowerSoil or PowerSoil-htp 96-Well DNA Isolation Kit (MoBio Laboratories, Carlsbad, CA, USA). Samples were incubated at 65 °C for 10 min and the manufacturer's protocol followed thereafter. Swab heads were treated using the same procedure, except that heads were snapped at the perforation into the extraction tube or block before starting the protocol, and the lysis buffer was added before the swabs were cut into the tubes and a shorter homogenization step was used. Wipe samples were stored in a sterile 250-mL tissue culture flask upon collection and thawed on ice before extraction. Cells were dislodged from wipes in a protocol adapted from Yamamoto et al.[13] Briefly, 150 mL of dislodging buffer (1× PBS, 0.04% Tween 80, passed through a 0.2-µm filter) was poured into a flask, and the flask was shaken vigorously for 1 min. The supernatant was then decanted into a 250-mL disposable filter funnel with a pore size of 0.2 µm (Thermo Scientific, Waltham, MA, USA) and the filter was then placed in a MoBio PowerWater extraction tube. After addition of PW1 buffer, the tubes were incubated at 65 °C for 5 min and the manufacturer's protocol followed thereafter. Illumina library construction followed standard protocols at the University of California QB3 Vincent J. Coates Genomics Sequencing Core Facility (http://qb3.berkeley.edu/gsl/) for S2_2013 and room_2013 samples. Briefly, genomic DNA was sheared using a Covaris to ~600 and 1000 bp. Wafergen's PrepX DNA library prep kits were used in conjunction with the Apollo324 robot following factory recommendations (Integenx). Thirteen cycles of PCR were used during library construction. Libraries were added in equimolar amounts, to the Illumina HiSeq platform. Paired-end sequences were obtained with 150 cycles and the data were processed with Casava version 1.8.2. Libraries for NIHY2_2012 samples were prepared using a previously described protocol[7].

**Assembly, binning, and read mapping**. Metagenomic sequencing of 140 fecal samples produced 0.5 Tb of Illumina HiSeq 2000 data for the NIHY2_2012 samples (Fig. 1). Subsequent metagenomic sequencing of 290 fecal samples produced 1.2 Tb of Illumina HiSeq 2500 data for S2_2013 samples. Metagenomic sequencing of 24 room samples produced 0.6 Tb of Illumina HiSeq 4000 data for the room_2013 samples. Reads were trimmed with Sickle[14], mapped to the human genome using Bowtie2[15] to remove human contamination, and assembled with idba_ud[16] using default parameters for all programs. Prodigal[17] was used in the "meta" mode for gene prediction of scaffolds longer than 1 kb. Genes were annotated using USEARCH[18] to search against KEGG[19], UniReff100[20], and UniProt databases. Matches with bit scores greater than 60 were saved as were reciprocal best hits with scores greater than 300. Ribosomal RNA sequences were identified using Infernal[21], and tRNAs with tRNAscan_SE[22]. Binning of fecal samples was conducted using the ggKbase binning interface. Binning of room samples was conducted using DAS Tool v1.0 using bins from CONCOCT v0.4.1[23], ABAWACA v1.00 (https://github.com/CK7/abawaca), and MaxBin v2.2[24] as input. To dereplicate genomes across time points and infants, dRep v0.3.3[9] was used with the secondary clustering threshold set at 99% gANI. To correct scaffolding errors, a reassembly script, ra2.py[25], was used on dereplicated genomes.

Mapping to detect strain presence or absence was done using a previously described workflow using PileupProfile.py[8]. Genomes with at least 10x coverage were considered in the analysis, and SNPs were called if 80% of reads disagreed with the reference base. In order to correct for SNPs resulting from nonspecific mapping or horizontal gene transfer events, SNPs within 100 bp of each other were not counted toward ANI calculation. Subspecies were considered present if genome breadth was > 90% and ANI > 99%, and strains were considered present if genome breadth was > 90% and ANI > 99.999%. Visualizations of genome bin overlap and mapping detection of these genomes was conducted in R[26] using ggplot2[27].

**K. pneumoniae strain shift**. Inspection of figures produced during infant S2_006 de-replication revealed the presence of multiple closely related strains of *K. pneumoniae*. Three 10-kb genome-specific regions were identified in each genome using Geneious v9.1.5, representing genome regions present in only one genome or the other. Infant S2_006 fecal samples were mapped to the set of genome-specific

regions, and the coverage of each fragment was used to estimate each strain's genome abundance.

**Data availability**. Genomes from assembled and binned room, and infant data, are available on NCBI under BioProjects PRJNA376580 and PRJNA376566, respectively. Short read data for room and infant samples were deposited in the Short Read Archive (accessions SRR5420274 to SRR5420297 and SRR5405607 to SRR5406014, respectively).

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

## Acknowledgements

Funding was provided through the Alfred P. Sloan Foundation under grant APSF-2012-10-05, NIH under grant 5R01AI092531, and the National Science Foundation's Graduate Research Fellowship Program to B.B. This work used the Vincent J. Coates Genomics Sequencing Laboratory at UC Berkeley, supported by NIH S10 OD018174 Instrumentation Grant. We thank C.T. Brown for his thoughtful critiques of the manuscript.

## Author contributions

J.F.B., M.J.M. and B.B. conceived the project. R.B. organized cohort recruitment and sample collections. B.A.F. conducted nucleic acid extractions and B.B. conducted the sample pooling. B.B. conducted the metagenomic assemblies and B.C.T. provided bioinformatics support. B.B. and M.R.O. conducted the strain-level read-mapping experiments. B.B. and J.F.B. wrote the final manuscript. All authors have read and approved the manuscript.

## Additional information

**Competing interests:** The authors declare no competing financial interests.

