## [Peer Review File · Nature Communications]

Reviewers' comments:

Reviewer #1 (Remarks to the Author):

Here the authors test the hypothesis that hospital room environment may contribute to microbiome development in preterm babies. They used whole genomes assembled from metagenomics data, they compared compare 317 bacterial genomes from the baby feces with 231 bacterial genomes from NICU surfaces, and found extensive sharing, with the most persistent being nosocomial infections-related aero-tolerant organisms (*Staphylococcus epidermidis*, *Enterococcus faecalis*, *Pseudomonas aeruginosa*, and *Klebsiella pneumoniae*). The results have important implications in the search for measurements that decrease preterm infections in hospital NICUs.

In this a well written manuscript, the results are very interesting an important, but there are some issues, particularly of clarification and acknowledgement of limitations of the study.

1. It seems that the results are based in 1 NICU and 6 babies. This needs to be explained, because the 50 babies studied in 2011-2014 and the 1038 samples of NICU falsely gives the impression of a big study.
2. How many years apart were the 6 babies separated?
3. Were all babies born vaginally?
4. The study lacks design for the time series collection of infant and NICU samples. This needs to be acknowledged or the design presented.
5. Not clear how are the 317 baby strains distributed among the 50 babies or among the 6 resampled babies. A Venn diagram would help.
6. In Fig 1: this is based on 3 babies and 1 NICU? Or infants were in different NICU rooms? Of these 6 babies, which shared or not the same room? How many rooms in total? .
7. The Cohort count (size of circles) is 10? Not 6?. Please clarify
8. From 448 strain genomes assembled in the study (317 from babies and 131 from NICUs), there were 12 shared between NICU and babies. But how many were found in the total 50 babies or in the 6 resampled babies?
9. How many NICUs? A single NICU in one hospital? Please clarify
10. How were the pre-term babies born? All were born vaginally?

On the other hand, there is a need of more discussion: what could be the sources of environmental bacteria? Speculate how is the environment selecting room reservoirs? Would it be affected by the presence of term babies? would a preterm baby's microbiome be different if sharing with a term baby? Or could it be affected by the presence of babies being breastfed? ... etc... or better to sterilize the room and request that no one enters exposing their skin?... pointing at what could be explored in the future, to reduce the burden of potential risks....

Reviewer #2 (Remarks to the Author):

Thank you for asking me to review this manuscript that attempts to address the potentially important issue of whether organisms really do stay in hospital environments and then colonise new babies.

The main concerns I have are around:

the pooling of the DNA for the environmental samples - from this manuscript alone I really can't tell how many samples were taken and pooled, over what time frame etc. As one of the key things the authors claim their data shows is that organisms really do stay in the environment this is crucial data, as pooling over a long time period could really compound this issue

I have tried very hard to understand the diagrams on sampling, and the authors have clearly tried very hard to put their complex methodology into a visual form, but I am afraid it was impossible to understand. I think the authors need to put into this paper more clear details around this very important aspect.

It is clearly possible that the babies colonise the environment not vice versa - again the data as presented did not to me justify why they feel that is definitely environment to baby rather than baby to environment

In addition as this is DNA the authors have not shown viability of the strains, simply presence.

Reviewer #3 (Remarks to the Author):

this manuscript intends to determine the direct link between environmental bacteria in NICU with the bacteria colonized in the preterm infants using metagenomic whole genome shotgun sequencing. it is an important contribution to the current understanding of the bacterial colonization in preterm infants.

Several issues need to be considered:

1. it is critical to define the similarity of the strains from rooms and infants. in the whole manuscript, different criterion was applied to define the strains are the same origin (L57,163,190). Can the author explain why not using the same standard?
2. from the total 50 infants, how many of them showed evidence of direct link between the infants bacteria and room bacteria? which body site (stool, skin or oral) is affected more by the environment?
3. can the authors comments how much influence of the room bacteria influences the infants' bacteria? from Extended fig1, only 12 strains were shared by room and infants, which seems a very small number.
4. it is known that there are inter-subjects strain variation, is there a between room variation too?
5. one caveat of the study is that although there is 99% identity between infant and room strains, it is hard to know how similar infants' strain with other potential sources such as caregiver, or mother. This make the *Staphylococcus epidermidis* conclusion is less solid.
6. the metadata of the infant is not provided.
7. there is no explanation of the meaning of each column in the suppl tables. it will be helpful to provide the info for certain key columns.
8. define 'bin'

Reviewer #1 (Remarks to the Author):

*Here the authors test the hypothesis that hospital room environment may contribute to microbiome development in preterm babies. They used whole genomes assembled from metagenomics data, they compared compare 317 bacterial genomes from the baby feces with 231 bacterial genomes from NICU surfaces, and found extensive sharing, with the most persistent being nosocomial infections-related aero-tolerant organisms (*Staphylococcus epidermidis*, *Enterococcus faecalis*, *Pseudomonas aeruginosa*, and *Klebsiella pneumoniae*). The results have important implications in the search for measurements that decrease preterm infections in hospital NICUs.*

We thank the reviewer for the positive evaluation.

In this a well written manuscript, the results are very interesting an important, but there are some issues, particularly of clarification and acknowledgement of limitations of the study.

Below we address each of you concerns and have incorporated most of your suggestions into the manuscript.

1. It seems that the results are based in 1 NICU and 6 babies. This needs to be explained, because the 50 babies studied in 2011-2014 and the 1038 samples of NICU falsely gives the impression of a big study.

We agree that the cohort information and metadata could be expanded to provide better understanding of the infants and samples involved in the study. To this end, we have added Supplementary Table 2, which includes specific SRA accessions for accessing short read data and many additional metadata fields (e.g., day of life, cohort, infant, birth day synced to study day, gestational age, sex, birth mode, birth weight, hospital, city, NICU room etc.). We think the additional metadata fields greatly improve clarity and accessibility to this study's supporting data.

To directly address your comment, the study was based on 50 infants (line 56) and 622 fecal metagenomes derived from new and previously published data (lines 55). These fecal samples are now detailed with metadata in the newly added Supplementary Table 2. The "1038 samples" reflects the number of room samples pooled in order to achieve enough biomass for deep shotgun sequencing. The details of these samples were in the initial submission, Supplementary Table 1.

2. How many years apart were the 6 babies separated?

Supplementary Fig.1 provides a temporal visualization of the infants' first day of life. Dots in the centrally placed timeline indicate birth dates of the 50 infants involved in this study. The collection period for all infants spans approximately 1100 days.

3. Were all babies born vaginally?

19 and 31 were born vaginally and caesarean section, respectively. Please see comment 1 and the newly added Supplementary Table 2 for details.

4. The study lacks design for the time series collection of infant and NICU samples. This needs to be acknowledged or the design presented.

Please see comment 1 and the newly added Supplementary Table 2 for details outlining time series collection.

All cohorts analyzed in this studied had time-series fecal collection. The added Supplementary Table 2 details fecal time series collection and Fig. 3 offers a focused perspective on temporal patterns of room/infant strain transfer.

5. Not clear how are the 317 baby strains distributed among the 50 babies or among the 6 resampled babies. A Venn diagram would help.

Supplementary Fig. 2, which is an expansion of main text Figure 1, details the distribution of subspecies across cohorts, infants, and rooms.

In our revised manuscript we conduct a second analysis with maximum stringency to improve confidence that two organisms are “the same”. We designate organisms whose sequences have $\geq 99.999\%$ average nucleotide identity (ANI) as “strains” and groups of sequences that share $\geq 99\%$ ANI as subspecies. This terminology is important in the responses that follow.

6. In Fig 1: this is based on 3 babies and 1 NICU? Or infants were in different NICU rooms? Of these 6 babies, which shared or not the same room? How many rooms in total? .

There were 14 unique rooms housing infants in the S2_2013 cohort. This room information has been added to Supplementary Table 2. Please see comment 1 and the newly added Supplementary Table 2 for details.

7. The Cohort count (size of circles) is 10? Not 6?. Please clarify

There were 6 room-infant pairs for which there is metagenomics data for both room and infant.

The dots indicate subspecies found within each cohort and the size of the dot indicates how many infants (or rooms in column 4) that subspecies was detected in. For better clarity, we have changed the size to reflect the percent of infants within each cohort that contain a particular strain.

8. From 448 strain genomes assembled in the study (317 from babies and 131 from NICUs), there were 12 shared between NICU and babies. But how many were found in the total 50 babies or in the 6 resampled babies?

75 of the 317 dereplicated infant fecal subspecies were found in > 1 infant (line 89). Details of which infants shared which subspecies is detailed in Supplementary Fig. 2.

9. How many NICUs? A single NICU in one hospital? Please clarify

The study was conducted in one NICU and each baby had its own room. Rooms and hospital columns have been added to the recently updated metadata. Additionally, a more detailed description of the NICU has been sited in the main body text (references 4, 9, 10, and 11). Please see comment 1 and the newly added Supplementary Table 2 for details.

10. How were the pre-term babies born? All were born vaginally?

Please see comment 3.

On the other hand, there is a need of more discussion: what could be the sources of environmental bacteria? Speculate how is the environment selecting room reservoirs? Would it be affected by the presence of term babies? would a preterm baby's microbiome be different if sharing with a term baby? Or could it be affected by the presence of babies being breastfed? ... etc... or better to sterilize the room and request that no one enters exposing their skin?.... pointing at what could be explored in the future, to reduce the burden of potential risks....

The article was originally submitted in Letter format, limiting the length in which we could extend discussion and speculation of the findings. While we could speculate on these topics, we have ongoing research to directly answer many of these questions and feel that these considerations are outside the scope of the current study. However, we have added substantial new analysis and figures. We have added discussion of the room reservoirs and briefly speculate that deeper sequencing of the room might have revealed greater overlap (this would point to strong infant-based selection).

Reviewer #2 (Remarks to the Author):

Thank you for asking me to review this manuscript that attempts to address the potentially important issue of whether organisms really do stay in hospital environments and then colonise new babies.

The main concerns I have are around:

the pooling of the DNA for the environmental samples - from this manuscript alone I really can't tell how many samples were taken and pooled, over what time frame etc.

As one of the key things the authors claim their data shows is that organisms really do stay in the environment this is crucial data, as pooling over a long time period could really compound this issue

I have tried very hard to understand the diagrams on sampling, and the authors have clearly tried very hard to put their complex methodology into a visual form, but I am afraid it was impossible to understand. I think the authors need to put into this paper more clear details around this very important aspect.

It is clearly possible that the babies colonise the environment not vice versa - again the data as presented did not to me justify why they feel that is definitely environment to baby rather than baby to environment

Similar to Reviewer 1's comments, we agree with both Reviewers that the experimental design was difficult to interpret. To provide clarity, we have added Supplementary Table 2, which includes specific SRA accessions for accessing short read data and many additional metadata fields (e.g. day of life, cohort, infant, birth day synced to study day, gestational age, sex, birth mode, birth weight, hospital, city, NICU room etc.). With respect to pooling room samples, this was provided originally in Supplementary Table 1. The additional metadata table greatly improves clarity and accessibility to this study's supporting data.

Concerning directionality, we agree with the Reviewer that microbial exchange is likely bidirectional. In lines 141-161 we note that of the twelve substrains in Infant 5, five were found in the room before detection in the infant. The remaining substrains were either detected in simultaneous time points or at later time points in the room. This suggests some are emitted from the infant to the room. The detection of these substrains long before their detection in the room (in infants from previous cohorts), suggests emission of strains from infant occupants to room surfaces may be the transmission route/cycle for these microbes.

In response to this question, we re-evaluated the criteria used to establish that strains in different reservoirs were the same. This greatly improved the robustness of the analysis of strain detection (prior to, simultaneously with, and following room occupancy).

In addition as this is DNA the authors have not shown viability of the strains, simply presence.

You are correct that we present no data to support viability in this paper. Replication rates for organisms colonizing infant 5's skin, oral, and fecal samples were recently published

(line 144, reference 8). In this study, replication rates were higher in skin and mouth samples compared to the gut. The replication rate analysis strengthens our inference based on repeated detection that there is a population of actively replicating cells.

Reviewer #3 (Remarks to the Author):

this manuscript intends to determine the direct link between environmental bacteria in NICU with the bacteria colonized in the preterm infants using metagenomic whole genome shotgun sequencing. it is an important contribution to the current understanding of the bacterial colonization in preterm infants.

We thank the reviewer for the positive evaluation.

Several issues need to be considered:

1. it is critical to define the similarity of the strains from rooms and infants. in the whole manuscript, different criterion was applied to define the strains are the same origin (L57,163,190). Can the author explain why not using the same standard?

Thank you for this very important comment. This comment, along with input from Reviewer 2, inspired us to re-evaluate the criteria used for comparative analyses. We have removed the figure that included information based on the “species-level” 96.5% threshold.

The revised analysis now has two components. In the first, we use the same 99% similarity that we calculated by the gANI algorithm as implemented in dRep using default parameters. This is comparable to stringency used in prior strain tracking analyses. We refer to these as subspecies (collections of exceedingly closely related strains) and note their relevance as potential colonizers of the infant microbiome.

The second, new analyses use the maximum stringency threshold of 99.999% (where maximum is defined based on comparison of reads and the matched genome sequence). This maximum stringency approach gives us very high confidence that the same strain was detected in different samples.

2. from the total 50 infants, how many of them showed evidence of direct link between the infants bacteria and room bacteria? which body site (stool, skin or oral) is affected more by the environment?

We directly address this question in the revised text (Line 107) and display the information.

“This analysis revealed 26 cases of identical strains present in both the room environment and infant gut (rows in Figure 4).”

Due to the limited number of skin and oral samples, we cannot attempt to answer the question of what site is most affected, but we hope our next sampling campaign can help provide clarity on this point.

3. can the authors comments how much influence of the room bacteria influences the infants' bacteria? from Extended fig1, only 12 strains were shared by room and infants, which seems a very small number.

The premature infant gut is a very low diversity environment so it is not surprising that the majority of room-associated strains were not detected in the infant gut. Importantly, however, the majority of the infant-associated organism types were found in the room.

4. it is known that there are inter-subjects strain variation, is there a between room variation too?

There are several cases where we found different strains of the same species in different rooms. The manuscript has been modified to clarify this. The information should now be apparent in Fig. 2.

5. one caveat of the study is that although there is 99% identity between infant and room strains, it is hard to know how similar infants' strain with other potential sources such as caregiver, or mother. This make the Staphylococcus epidermidis conclusion is less solid.

As noted in responses to the other reviewers, this and other reviewer comments motivated the addition of a maximum stringency analysis (at the ≥ 99.999 % ANI level) that allows us to discriminate extremely similar strains from each other and provide high confidence as to potential sources.

We agree that we cannot rule out human activity and have included this comment in the revised manuscript.

6. the metadata of the infant is not provided.

Please see the newly added Supplementary Table 2 for metadata details. The new table contains specific SRA accessions for accessing short read data and many additional metadata fields (e.g. day of life, cohort, infant, birth day synced to study day, gestational age, sex, birth mode, birth weight, hospital, city, NICU room etc.). We think the additional metadata fields greatly improve clarity and accessibility to this study's supporting data.

7. there is no explanation of the meaning of each column in the suppl tables. it will be helpful to provide the info for certain key columns.

Supplementary Tables (2-11, Extended Data 1-9) were generated using dRep (line 83). dRep provides extensive documentation to help readers interpret the output. For tables not generated with dRep we have added a comment line that provides description for each column header.

8. define 'bin'

Bins are generated using the methods and software sited in lines 204-220.

REVIEWERS' COMMENTS:

Reviewer #1 (Remarks to the Author):

The authors have provided satisfactory responses to criticisms and suggestions, and modified the manuscript accordingly.

Reviewer #2 (Remarks to the Author):

thank you for the responses to the issues raised
the manuscript is clearer in some areas, but still despite my best efforts I do not understand important points of methodology

I suspect that to the authors it is very clear, but after significant time perusing graphs and figures, reading and re-reading the manuscript I still do not understand the room sampling and analysis and how these pooled samples relate to timing of a stool sample.

My specific issues that I do not understand are:

From a room, how frequently were the 3 sample types obtained, and what time period was pooled. I simply cannot see this anywhere - so I would like (for example) that samples were collected daily from all 3 sites, and 3 months worth had to be pooled to get enough DNA (or whatever the truth is)

As the room occupants change, this pooling of samples is crucial to understanding quite what the message is

Most readers will not engage with having to hunt for this (and I have hunted and still can't see it)

Why the assumption is that where the strain is seen in the room and the infant, that the direction of travel is room to infant, when it is surely more likely the other way, or that both get colonized from an intermediary (parents/staff etc)

More discussion is needed about the fact that infants are (virtually) sterile at birth and HAVE to get colonized from somewhere with something - if you are in a NICU and not allowed much skin to skin contact with parents then the 'only' source is your environment including the staff that care for you and the food you are given.

Reviewer #3 (Remarks to the Author):

The revised version is improved tremendously. suggest for publication.

REVIEWERS' COMMENTS:

Reviewer #1 (Remarks to the Author):

The authors have provided satisfactory responses to criticisms and suggestions, and modified the manuscript accordingly.

We thank the reviewer for the positive evaluation.

Reviewer #2 (Remarks to the Author):

*thank you for the responses to the issues raised
the manuscript is clearer in some areas, but still despite my best efforts I do not understand
important points of methodology*

*I suspect that to the authors it is very clear, but after significant time perusing graphs and
figures, reading and re-reading the manuscript I still do not understand the room sampling and
analysis and how these pooled samples relate to timing of a stool sample.*

My specific issues that I do not understand are:

*From a room, how frequently were the 3 sample types obtained, and what time period was
pooled. I simply cannot see this anywhere - so I would like (for example) that samples were
collected daily from all 3 sites, and 3 months worth had to be pooled to get enough DNA (or
whatever the truth is)*

*As the room occupants change, this pooling of samples is crucial to understanding quite what the
message is*

Most readers will not engage with having to hunt for this (and I have hunted and still can't see it)

After the first round of submission, we agree this information was difficult to find. Upon resubmission, however, we included a sample pooling metadata table. Within a few clicks, one can find how samples were pooled. An example is below. If you filter columns for Infant 3 and only filter for the “sinks” pool, you can see that 13 samples were pooled spanning day of life 5 to 21. This information can also be linked to infant metadata in Supplementary Data 2.

	id	infant	dol	env	pool	room_mg	cohort	cohort_infant
5	0305F		3	5 sink_basin	sinks	S2_003_000_R2	S2_2013	Sloan2_3
7	0306F		3	6 sink_basin	sinks	S2_003_000_R2	S2_2013	Sloan2_3
8	0307F		3	7 sink_basin	sinks	S2_003_000_R2	S2_2013	Sloan2_3
9	0308F		3	8 sink_basin	sinks	S2_003_000_R2	S2_2013	Sloan2_3
10	0311F		3	11 sink_basin	sinks	S2_003_000_R2	S2_2013	Sloan2_3
11	0312F		3	12 sink_basin	sinks	S2_003_000_R2	S2_2013	Sloan2_3
12	0313F		3	13 sink_basin	sinks	S2_003_000_R2	S2_2013	Sloan2_3
13	0314F		3	14 sink_basin	sinks	S2_003_000_R2	S2_2013	Sloan2_3
14	0315F		3	15 sink_basin	sinks	S2_003_000_R2	S2_2013	Sloan2_3
15	0318F		3	18 sink_basin	sinks	S2_003_000_R2	S2_2013	Sloan2_3
16	0319F		3	19 sink_basin	sinks	S2_003_000_R2	S2_2013	Sloan2_3
17	0320F		3	20 sink_basin	sinks	S2_003_000_R2	S2_2013	Sloan2_3
18	0321F		3	21 sink_basin	sinks	S2_003_000_R2	S2_2013	Sloan2_3

Why the assumption is that where the strain is seen in the room and the infant, that the direction of travel is room to infant, when it is surely more likely the other way, or that both get colonized from an intermediary (parents/staff etc)

We agree that directionality will largely be from infant to room, in terms of absolute cell numbers. However, we feel the novel aspect of this work is that once in the room, microbes can tolerate room conditions until gut colonization of a downstream occupant is possible. We highlight both directional transfers within the manuscript (e.g. infant to room on line 141 and room to infant on line 140). We also agree that caretakers are likely vehicles and raised this possibility on line 147.

More discussion is needed about the fact that infants are (virtually) sterile at birth and HAVE to get colonized from somewhere with something - if you are in a NICU and not allowed much skin to skin contact with parents then the 'only' source is your environment including the staff that care for you and the food you are given.

We agree that infants housed in a NICU, many receiving antibiotic treatment during the first days of life, likely source microbial colonists from their immediate environment. On lines 20 and 91, we reference publication that have described the rationale for studying NICU housed preterm infants.

Reviewer #3 (Remarks to the Author):

The revised version is improved tremendously. suggest for publication.

We thank the reviewer for the positive evaluation.

Thank you for the review of our manuscript and for allowing us the opportunity to revise and respond to reviewer comments. We have addressed each reviewer's comments on a point-by-point basis and incorporated a number of positive changes to the submitted revised manuscript.